# Is Satellite Ahead of Terrestrial in Deploying NOMA for Massive Machine-Type Communications?

**DOI:** 10.3390/s21134290

**Published:** 2021-06-23

**Authors:** Antonio Arcidiacono, Daniele Finocchiaro, Riccardo De Gaudenzi, Oscar del Rio-Herrero, Stefano Cioni, Marco Andrenacci, Riccardo Andreotti

**Affiliations:** 1European Broadcasting Union (EBU), L’Ancienne-Route 17A, 1218 Le Grand-Saconnex, Switzerland; arcidiacono@ebu.ch; 2Eutelsat, 32 boulevard Gallieni, 92130 Issy-les-Moulineaux, France; dfinocchiaro@eutelsat.com; 3European Space Agency (ESA), European Space Research and Technology Centre, Keplerlaan 1, P.O. Box 299, 2200 AG Noordwijk, The Netherlands; Oscar.del.Rio.Herrero@esa.int (O.d.R.-H.); Stefano.Cioni@esa.int (S.C.); 4MBI SrL, Via Francesco Squartini, 7, 56121 Pisa, Italy; mandrenacci@mbigroup.it (M.A.); randreotti@mbigroup.it (R.A.)

**Keywords:** satellite communications, massive machine type communications, Internet of Things, non orthogonal multiple access, random access

## Abstract

Non-orthogonal multiple access (NOMA) technologies are considered key technologies for terrestrial 5G massive machine-type communications (mMTC) applications. It is less known that NOMA techniques were pioneered about ten years ago in the satellite domain to match the growing demand for mMTC services. This paper presents the key features of the first NOMA-based satellite network, presenting not only the underlying technical solutions and measured performance but also the related deployment over the Eutelsat satellite fleet. In particular, we describe the specific ground segment developments for the user terminals and the gateway station. It is shown that the developed solution, based on an Enhanced Spread ALOHA random access technique, achieves an unprecedented throughput, scalability and service cost and is well matched to several mMTC satellite use cases. The ongoing R&D lines covering both the ground segment capabilities enhancement and the extension to satellite on-board packet demodulation are also outlined. These pioneering NOMA satellite technology developments and in-the-field deployments open up the possibility of developing and exploiting 5G mMTC satellite- and terrestrial-based systems in a synergic and interoperable architecture.

## 1. Introduction

Non-orthogonal multiple access (NOMA)-based systems have been recently investigated by the 3rd Generation Partnership Project (3GPP) [1] as a promising set of emerging technologies able to provide a more efficient utilization of wireless resources for future 5G networks. Research activities have been accelerating in recent years around these technologies but are mainly confined to theoretical and simulation analysis for terrestrial wireless applications.

Ten years before the 3GPP 5G standardization effort started, NOMA technologies were pioneered for Internet of Things (IoT) applications to exploit, at best, the limited resources of satellite-based networks, starting from mobile satellite service (MSS) applications below 3 GHz. The requirements set forth for MSS systems were as follows:Efficiently and reliably support a very large number of users, sporadically transmitting small to medium-sized packets, typical of satellite-based IoT applications;Capable of operating in systems with a limited channelization bandwidth per service area (e.g., from 0.2 to a few megahertz);Energy-efficient solution allowing unattended terminal operation for a long time;Easy network scalability, overhead minimization and low-cost, easy to install terminals.

These challenging requirements stimulated the search for an appropriate solution, and the result was the development of the first NOMA-based system using an Enhanced Spread Spectrum ALOHA (E-SSA) [2]. Random access (RA) scheme, featuring iterative successive interference cancellation (i-SIC), implemented, for the first time, in the S-band MSS frequency range [3]. The solution was also standardized by the European Telecommunications Standards Institute (ETSI) as S-band Mobile Interactive Multimedia (S-MIM) [4].

Once the solution developed for those relatively low frequency bands demonstrated its excellent efficiency and flexibility, it was decided to extend the use of this NOMA-based system to higher frequency bands. The S-MIM extension targeted the Ku and Ka bands (corresponding to 11–14 GHz and 20–30 GHz frequency ranges, respectively), where a few gigahertz of bandwidth per service area was available and commercially exploited by a large number of geostationary satellites. This development effort has materialized into the specification and implementation of the so-called F-SIM system [5] and the launch of the Eutelsat SmartLNB technology, today operationally deployed in four continents under the “IOT FIRST” brand, with SmartLNB terminals reaching their third generation [6].

For the first time, a satellite NOMA-based system has been conceived, developed, industrialized and put into operation, and hence several lessons can be derived from this experience and used to better guide future terrestrial wireless-related developments. This is opening up the possibility of developing and exploiting satellite- and terrestrial-based systems in a synergic and interoperable architecture.

The renewed interest in 3GPP in the integration between satellite-based and terrestrial-based 5G systems is a good precursor to the integration of a NOMA-based multiple access system, starting from a proposal for 3GPP to be integrated in release 18 of the 5G standard. A proposal inspired by the long development and operational experience cumulated in the last 10 years, in the actual implementation of E-SSA-based systems, may represent a solid premise for an effective integration of satellite and terrestrial systems for massive machine-type communications (mMTC).

The paper is organized as follows: in Section 2, the NOMA-based S-MIM system is described; in Section 3, the evolution from S-MIM to F-SIM is summarized jointly with key performance laboratory results; Section 4 provides an overview of the F-SIM system ground segment elements developed; Section 5 shows the ongoing R&D activities aiming to further improve the performance and to expand the technology use cases; Section 6 discusses the possible satellite technology commonalities with 5G eMTC requirements; finally, Section 7 provides the conclusions.

## 2. The S-MIM System

### 2.1. Historic Background

A few years after the turn of the millennium, the European Commission (EC) accepted the satellite industry proposal to bring to use some spectrum in the S band, the mobile satellite service (MSS) band, ranging between 1980–2010 and 2170–2200 MHz [7]. The frequency allocation also allowed the deployment of iso-frequency terrestrial gap fillers to ensure high-quality service provision in urban and suburban areas. The satellite band allocation was conveniently adjacent to the terrestrial third-generation Universal Mobile Telecommunication System (UMTS) allocated band, thus making it possible to exploit synergies between satellite and terrestrial UMTS services. The main difficulty was that, in order to support mobility, the user antenna had to be very small—and this implied the need for a large deployable antenna reflector on the satellite, which was at the limit of what industry could provide.

The EC decided to split the available bandwidth into two slots (each comprising 15 MHz for the downlink and 15 MHz for the uplink) to be operated over the whole European Union by two different entities. After a competitive selection process, Solaris Mobile Limited and Inmarsat Ventures Limited were assigned the S-band MSS spectrum.

The main use case scenarios identified for the MSS were as follows:Broadcasting multimedia content to handheld user terminals, in the spirit of the XMRadio/Sirius experience in the US, but extended to encompass video content broadcasting;Mobile data acquisition services, such as collecting data from mobile sensors (e.g., vehicles), toll payments and environmental monitoring—today considered as part of the “Machine to Machine” (M2M) or “Internet of Things” (IoT) services.

The two use cases were perfectly complementary, as each used only one direction of the satellite path (forward link only for the multimedia broadcasting, and almost exclusively the return link for mobile data acquisition/IoT). IoT applications have a great market potential, as the number of “connected objects” is expected to have an exponential growth in the coming years. Consequently, the satellite, complemented by the ancillary terrestrial gap fillers network, could capture this new market quickly, providing pan-European coverage. At the same time, terrestrial networks were not yet prepared to support mMTC, and their planned coverage was concentrated in populated areas. The satellite had the advantage of full global coverage, including unpopulated areas where mobile users or objects should be served (e.g., for environmental monitoring).

Some stringent requirements on the selected communication protocol had to be satisfied in order to serve the mobile IoT market from the satellite:Reliable performance when operating in typical land mobile satellite (LMS) channels;Massive scalability, i.e., capability to handle a very large number of objects (order of millions);High spectral efficiency, as the available spectrum was limited. It was necessary to accommodate all the capacity requests of a spot beam (roughly the size of a European country such as Italy or Germany) in the 5 MHz allocated per beam, and to provide a cost per bit appropriate for the IoT market;Low-cost technology for the objects: as the typical sensor costs just a few dollars, the communication part should be of the same order of cost. This implies, in particular, a limited transmit power, simple algorithms and loose requirements on clock synchronization;Optimized for small transactions, typical of objects communication, minimizing overheads such as IP headers or bandwidth assignment demands.

As the search for an optimal protocol for the satellite broadcast application had already been successfully concluded by means of the standardized DVB-SH protocol [8,9,10], the effort towards the satellite IoT protocol was concentrated on the return link only, and the future ETSI S-MIM standard would be a combination of DVB-SH and the newly defined E-SSA return link protocol.

### 2.2. Selected NOMA Solution

Stimulated by the challenging requirements for the IoT use case, the European Space Agency (ESA) initiated investigations of advanced random access (RA) techniques able to satisfy the satellite operator’s needs. Previous research on slotted RA solutions with collision resolution was not retained. The main drawback of slotted RA is related to the need to keep the terminal time synchronized, causing an unwanted level of signaling overhead, and increasing the terminal complexity. A comprehensive survey of satellite RA schemes is provided in [11].

For the new IoT protocol, the attention was rapidly turning to the E-SSA NOMA technology, as it was providing the best fit to the requirements listed in Section 2. In particular, E-SSA allows full asynchronous uncoordinated access with high packet delivery reliability, a low transmit peak power and high energy and spectral efficiency (three orders of magnitude higher than classical ALOHA [12]). In addition, the S-MIM standard provides a simple yet effective open loop power and packet transmission control technique, maximizing the successful transmission of packets, even in the presence of satellite mobile channel blockage and shadowing.

E-SSA represents an evolution of the well-known Spread Spectrum ALOHA (SSA) random access protocol proposed by Abramson [12]. The main difference between E-SSA and SSA is related to the gateway demodulator processing. Instead of a single-packet demodulation/decoding SIC attempt, the E-SSA demodulation processing is based on a sliding window approach. The sliding window I–Q baseband signal, typically spanning the length of three packets, is repeatedly scanned, searching for detectable packets performing iterative SIC. The packet detection is based on a known preamble. Under loaded conditions, during the first SIC pass, only a subset of packets’ preambles can be detected due to the high level of multiple access interference. However, this relatively small percentage of detected packets, if successfully decoded, is then reconstructed at the baseband and subtracted from the demodulator sliding window memory. To obtain an accurate baseband packet reconstruction, the full detected payload is used to perform a decision-directed channel estimation. Once the first pass and associated detection and successive interference cancellation (SIC) step is completed, the process is restarted, and more packets can be detected and demodulated thanks to the previous cancellation step. This process is repeated a number of times (iterations) until all packets are recovered. Then, the observation window (typically the length of three packets) is shifted by a fraction (e.g., 1/3) of the packet duration, and the iSIC process repeated [2].

The E-SSA analysis and simulation results reported in [2] have shown that the packet loss ratio (PLR) rapidly falls to low values below a critical medium access control channel (MAC) normalized load (expressed in bits/chip). The throughput and PLR behavior of SSA and E-SSA with and without a packet power unbalance are reported in Figure 1. In particular, Figure 1a shows the normalized throughput expressed in bits/chip vs. the average medium access channel (MAC) normalized load, also expressed in bits/chip for both conventional SSA and E-SSA. Figure 1b provides the packet loss ratio (PLR) vs. the normalized MAC load. Results are obtained for the balanced and unbalanced power of the received packets. A lognormal packet power distribution is assumed with standard deviation of σ = 0 dB (balanced power) and σ = 3 dB (unbalanced power). Note that in the case of the power unbalance, the SSA throughput is heavily impacted by the well-known CDMA near–far effect, while the E-SSA performance is further boosted thanks to the SIC processing. The E-SSA PLR horizontal floor appearance for the lognormal packet power distribution is explained in [2], and it is fully predictable. For an IoT satellite-based system, it is important to maximize the first packet transmission successful reception at the gateway, in order to minimize the number of retransmissions for energy efficiency and latency reasons. For this reason, a reasonable target packet error rate (PER) is 10^−3^ or less. Assuming this typical target PER, we observe that in the presence of a packet power unbalance, the E-SSA throughput is several orders of magnitude higher than SSA. The steep PLR vs. MAC load E-SSA characteristic also allows simplifying the congestion control as it is sufficient to keep the MAC average load below a certain critical value, which can be easily monitored at the gateway measuring the interference plus thermal noise over thermal noise ratio.

Another interesting feature of E-SSA is that it can be operated with a single preamble despite the non-zero probability of preamble collision. This is because the asynchronous random access nature (i.e., random delay of arriving packets) combined with the carrier frequency uncertainty due to the terminal oscillator instabilities, as well as the possible differential Doppler, makes the preamble destructive collision probability low enough not to require multiple preambles. This makes the E-SSA demodulator implementation easier, limiting the need for a single preamble searcher. As a matter of fact, the preamble detection function is the most demanding demodulator functionality in terms of processing requirements.

## 3. From S-MIM to F-SIM

In the light of the good performance provided by S-MIM for mobile applications, it was a natural decision to extend it to other use cases, particularly for the case of fixed terminals using legacy GEO satellites in the C, Ku or Ka band. The objective was to offer cheaper VSAT-like services with a technology that would allow reducing both the terminal cost and service cost, more appropriate for applications such as interactivity, or IoT backhauling, where broadband speeds are not necessary. The new protocol, derived from S-MIM, was named the Fixed Interactive Multimedia Services (F-SIM) protocol. F-SIM was adapted from S-MIM specifically in order to efficiently support fixed terminal operations. As detailed in [5], the main differences are as follows:(a)Support of higher frequency satellite bands: C, Ku and Ka—including adapted uplink power control algorithm in order to support propagation channel characteristics in these bands;(b)Use of digital video broadcasting DVB-S2/S2X protocol (instead of DVB-SH) in the forward link, including a network clock reference (NCR) counter, allowing the terminal to achieve an accurate frequency reference;(c)New physical layer configurations, in terms of bit rate and packet size;(d)Native support of Internet Protocol (IP) and flexible management of the different quality of service (QoS) classes at data link layer.

F-SIM was designed in order to enable these two classes of services:“Satellite Over the Top” services: additional interactive IP-based services on top of video satellite broadcasting, e.g., video to personal devices (multiscreen), digital rights management (DRM), voting, real-time audience measurement, targeted advertising, limited web browsing and datacast;IoT/M2M connectivity: message-based or low-bit rate connectivity for objects or small networks. This includes IoT services, supervisory control and data acquisition (SCADA) and backhauling of terrestrial low-power wide-area networks (LPWAN).

### 3.1. F-SIM Physical Layer

The E-SSA waveform specification adopted in S-MIM and F-SIM is based on the 3GPP W-CDMA uplink waveform [13,14,15]. The payload is carried by the physical layer data channel (PDCH), while pilot symbols for unknown parameter estimation (e.g., time, frequency and phase) at the receiver and an optional signaling field providing information on the actual carrier format are carried by the physical layer control channel (PCCH). The same forward error correction (FEC) scheme (Turbo code with rate 1/3) and BPSK modulation are adopted on both channels. Similar to 3GPP W-CDMA, the two channels are spread and mapped to the I and Q components of a complex signal which is, in turn, scrambled by a complex long spreading code.

As described in [5], differently from S-MIM, F-SIM defines four possible channel sizes: 2.5, 5, 10 and 40 MHz (the bandwidth actually occupied is, respectively, 2.34, 4.68, 9.36 and 37.44 MHz, with a 22% square root-raised-cosine chip shaping filter roll-off factor). Different spreading factors are defined, ranging from 16 to 256, in order to adapt to the link budget and the required data rate. Finally, different packet sizes are defined, from 38 to 1513 bytes, in order to minimize the amount of padding bytes in each packet sent, and to optimize the use of the bandwidth.

As an example, for channelization of 10 MHz and a spreading factor of SF = 16, the burst duration varies from 2 (38 byte payload) to 75 ms (1513 byte payload), and the minimum C/N for reception is −15.2 dB (at PER = 10^−4^). For the same channelization and a spreading factor of SF = 256, the burst duration varies from 32 to 240 ms (300 byte payload), and the minimum C/N for reception is −27.3 dB (at PER = 10^−4^).

Similar to S-MIM, F-SIM uses dual-BPSK modulation with the FEC coding rate 1/3, as shown in Figure 2. The Turbo code from 3GPP Release 99 specifications has been adopted [13].

The uplink burst composition is depicted in Figure 3 and further described hereafter. The PDCH carries the random access channel (RACH) data burst followed by a CRC. The PCCH carries physical layer signaling and reference symbols to allow coherent demodulation of the PDCH channel. The physical layer signaling conveys the transport format indication (TFI), particularly information associated with the spreading factor and data burst length used in the modulation process. Each channel is scrambled with an orthogonal variable spreading factor (OVSF) code, and a final scrambling is performed using Gold codes. Different scrambling codes can be used in the system, but, in general, a single scrambling code for each satellite beam is used. The preamble is composed of a sequence of 96 symbols, spread by a pseudorandom noise (PN) code of the period and spreading factor equal to the spreading factor used for the current packet.

With the F-SIM physical layer being very close to the terrestrial 3GPP W-CDMA return link standard, there are also benefits from its suitability to be embedded in low-cost handheld user terminals. One of the W-CDMA uplink features is to have a moderate peak-to-average envelope fluctuation (see Figure 3 in [16]), as it is based on a power unbalanced dual-BPSK modulation, resulting in an asymmetric 8PSK constellation (see Figure 2 in [16]). This, combined with the use of low-code rate FEC, makes F-SIM suitable for driving the user terminal solid-state power amplifier (SSPA) in a moderately compressed mode (i.e., 2–3 dB compression mode) with no appreciable performance degradation. It should be remarked that when link margins are available, it is preferable to randomize the transmit power to achieve a higher throughput [17]. This power randomization approach has been implemented in F-SIM-based networks.

For what concerns the satellite return link transponder nonlinearity effects, they are normally negligible as the transponder operates in the multi-carrier mode with many co-frequency spread spectrum carriers on each F-SIM dedicated frequency slot, and multiple frequency slots dedicated to F-SIM or other services. In this case, the transponder high-power amplifier (HPA) is operated in a moderately linear mode (e.g., >4 dB of output back-off) to minimize the HPA’s intermodulation effects as it is common practice in any multi-carrier satellite transponder.

### 3.2. F-SIM Link Layer

The F-SIM link layer specifies the state machine of the modem. When there are no data to transmit, the terminal transmit chain is completely off, thus saving on power consumption. The terminal logs into the network with a single transmission (logon request) which is valid for hours. The hub continuously transmits signaling information, which is shared by all terminals, in order to support a large number of terminals. F-SIM is designed to transport native Internet Protocol (IP) packets. The link layer on the terminal side encapsulates each IP packet into one or more fragments according to its length. Each fragment is sent separately, with a minimal encapsulation. On the hub side, fragments are reassembled after demodulation, and the resulting IP packet is routed according to normal IP routing policies. The specifications support the deployment of separate IP address spaces, i.e., each customer of the platform can freely use the entire address space, with their own routing rules, without any conflict with other customers.

Two important tasks are performed by the link layer:Power spreading optimization—The overall throughput of the system is optimized when packets are received at the hub with different power levels, ideally a uniform distribution if expressed in dBm. The algorithms in the F-SIM link layer therefore randomly adjust the outgoing packet power, within the available link margin, to ensure this property [17].Congestion control—when the system approaches saturation, signaling information is generated by the hub and used by the terminals to slow down or stop transmission for low-priority services.

One interesting feature of F-SIM is that different spreading factors are supported for the same channelization, which allows adjusting the uplink speed to the current link conditions. The terminal continuously monitors the forward link received power level, in order to compute the expected return link budget. When a packet is ready to be sent, the most appropriate spreading factor is selected, e.g., maximizing speed while guaranteeing that the link budget closes.

Spreading factors can also be allocated statically to certain services. In fact, the overall network performance is increased if many terminals use a large spreading factor. Therefore, if the uplink speed is not important for a certain use case, a service could be allocated to a higher spreading factor.

### 3.3. F-SIM Forward Link (DVB-S2)

The F-SIM specifications do not force the use of a specific forward link but only require that it supports certain features. In practice, the forward link is implemented with the DVB-S2 protocol, using MPE encapsulation to support the IP. Signaling is transported as multicast IP streams containing compressed Protobuf structures. DVB adaptive coding and modulation (ACM) is supported, with the following advantages:Supporting terminals with different performances, e.g., different antenna sizes;Providing high availability, by using strong DVB-S2 physical layer configurations (MODCODs) when required by weather conditions;Increase coverage up to the limits of the satellite beam;Avoiding bandwidth waste, by switching to more efficient MODCODs if the link budget permits this action.

Extensions to DVB-S2X very low signal-to-noise ratio (VL-SNR) modes have been tested, as well as the use of Generic Stream Encapsulation (GSE). The DVB-S2 signal also contains an NCR (network clock reference) counter running at 27 MHz, in order to help the terminals to correct the local clock, in order to transmit signals with a very accurate frequency and symbol rate. Alternatively, the terminals can use the symbol rate of the forward link as the frequency reference.

### 3.4. F-SIM Key Implementation Aspects and Laboratory Test Results

This section summarizes some key demodulator implementation aspects and the laboratory test results for some F-SIM waveforms previously introduced, corresponding to different use cases of interest. Performance results are shown in terms of the aggregated throughput (bits/chip) vs. the average multiple access channel (MAC) offered load (bit/chip) and the packet loss ratio (PLR) vs. the average MAC offered load (bit/chip).

One of the main challenges in achieving the excellent theoretical E-SSA performance is related to the gateway demodulator’s ability to minimize implementation losses in the presence of user terminal carrier frequency errors and phase noise. In particular, the two most critical demodulator functions are as follows:-Packet preamble acquisition;-Interference cancellation.

As mentioned at the end of Section 2.2, the first demodulation block is the preamble searcher (PS). The PS detects bursts in the current sliding window, by performing a search over a time–frequency grid to find the timestamp (i.e., the sample corresponding to the beginning of a burst) and the coarse frequency offset estimation of the detected bursts. Such a search is carried out by performing a cross-correlation between the samples stored in the sliding window memory and a local replica of the preamble (efficiently carried out by means of the fast Fourier transform). The spacing between the frequency hypothesis is set to keep the correlation loss [18] very low (usually, 0.5 dB), given the coherent integration time. This value corresponds to the duration of a preamble symbol, and then the resulting $N_{p}$ coherent integrations (one for each symbol of the preamble, whose length is $N_{p}$ symbols) are combined non-coherently. Such a short coherent integration allows having a rather wide spacing between the frequency hypotheses, or, in other words, to have a limited and hence affordable number of frequency hypotheses to test given the maximum frequency offset a burst can be received with. This maximum value accounts for all the impairments the carrier frequency is subject to, mainly, the instability of the terminal local oscillator. Thus, the PS provides the time delay estimate and a first coarse frequency offset estimate. Then, the residual frequency offset is finely estimated by a data-aided frequency estimation algorithm, i.e., the Rife–Boorstyn (RB) algorithm working on the known preamble symbols and pilot symbols [19]. Finally, the phase of the received signal is estimated. This is carried out by the maximum likelihood phase estimation algorithm working on the pilot symbols within a sliding window which has a length of M pilot symbols and slides of K=1 symbols at a time. The length M is tuned to have a good trade-off between the estimation accuracy and tracking of the phase variation due to the effect of the phase noise. To summarize, the combination of the several estimators described above allows the demodulator to be robust to the channel impairments, thus justifying the limited performance loss shown in the results.

Concerning the detected packet cancellation, it is well known that the SIC process can be negatively impacted in the presence of an imperfect cancellation caused by channel estimation errors (carrier frequency, phase, amplitude and clock timing). Once a burst is successfully decoded, the physical burst is locally regenerated so that a refined channel estimation is carried out, exploiting all the burst payload symbols in a decision-directed mode. This allows a better estimate of the carrier amplitude and phase evolution over the packet, thus leading to a better cancellation from the sliding window memory. Then, the locally regenerated and corrected burst is subtracted from the sliding window memory. To summarize, the combination of the several estimators described above allows the demodulator to be robust to the channel impairments, thus justifying the limited performance loss.

The first use case (UC#1), named High Efficiency F-SIM, aims at showing the F-SIM performance in a typical GEO Ku band scenario with a SmartLNB terminal (0.5 W RF power, 75 cm dish). F-SIM waveforms Cr3840Sf16Ds38 and F-SIM Cr3840Sf64Ds38 [20] (i.e., with chip rate of $R_{c}=3.84$ Mchip/s, data size equal to 38 bytes and two spreading factors SF = 16 and SF = 64) were simulated. Link budgets with typical Ku-band satellite ratings show a link margin at the center of coverage (that can be used for power spreading) of about 4 dB for the selected waveform when SF = 16 is employed with a minimum C/N = −14 dB, and of about 10 dB for SF = 64 with a minimum C/N = −20 dB. Thus, with SF = 64, power randomization of 9 dB is considered. Figure 4 and Figure 5 show the throughput and PLR performance, respectively, also considering the case of phase noise at the terminal, with the phase noise (PN) mask as described in [20]. The number of SIC iterations was set to a high value (up to 32) to obtain the maximum performance, although for a realistic performance, it could be lower. Results are also summarized in Table 1. The increased throughput (factor 2 increase) achievable, passing from a spreading factor of 16 to 64, can be observed. We also remark that the phase noise effect on the performance is quite limited thanks to the robust gateway demodulator processing algorithms, introduced at the end of Section 2.2.

## 4. Developed E-SSA-Based NOMA System Elements

For mobile broadcasting IoT applications, on the industrial side in 2008, Eutelsat and SES satellite operators joined their forces in the Solaris joint venture, which was in charge of developing the hybrid system and procuring the S-band payload of the Eutelsat W2A satellite. The payload included, in particular, a multi-port amplifier (MPA) allowing a flexible allocation of power among the six linguistic-shaped beams, and a 12 m reflector antenna capable of delivering typically an EIRP of ~60 dBW and a G/T of ~10 dB/K on each beam. The W2A satellite depicted in Figure 6 was launched in 2009, but a problem related to incorrect antenna deployment rendered the commercial development of the foreseen S-band services impossible in practice. Despite the satellite non-nominal antenna performance, extensive field trials were performed in France, Spain and Italy.

The S-MIM and F-SIM ground segment technologies have been developed by different companies, under the coordination of Eutelsat.

In particular, in the following, we focus on the most recent F-SIM ground equipment:MBI (Italy) has developed the gateway, implementing, in particular, the E-SSA packet demodulator, and integrated it into a commercial hub product dubbed “HyperCube”.Enensys (France) has improved its “SmartGate” DVB-S2 modulator to fully support the F-SIM forward link.Egatel (Spain) and Ayecka (Israel) have developed fixed terminals, named “SmartLNB”, to be used with DTH-like parabolic dishes (75–80 cm diameter).The terminals have also been integrated into auto-pointing nomadic antennas, as well as maritime antennas from Intellian (US) and KNS (Korea).Work is in progress to integrate the F-SIM modem into a flat antenna with electronic steering from Satixfy (Israel).

### 4.1. Fixed Terminals

The F-SIM protocol has been integrated into the innovative all-in-one terminal concept named “SmartLNB”. In collaboration with Eutelsat, two manufacturers (Egatel and Ayecka) have conceived and produced terminals that are now available as commercial products (see Figure 7), with a lower price with respect to competitive two-way satellite terminals. The user terminal key technical characteristics are as follows:Integrated modem and low-noise block (LNB)/block upconverter (BUC) into an integrated outdoor unit (ODU) design (15 × 11 × 3 cm, 1.3 kg);Coaxial or Ethernet connection from ODU to indoor unit (IDU) with data and power supply sharing the same cable;Ku-band linear polarization;Maximum transmit output power: 27 dBm, resulting in 36 dBW effective isotropic radiated power (EIRP) with a typical 70 cm dish;Support of all DVB-S2X MODCODs;Modem based on the ST Cardiff3 chipset;Power consumption: 0.5 W (standby), 7 W (receive only), 16 W (receive and transmit);Based on Linux operating system, and supporting TCP/IP, VLAN, VRF, IPSEC and DHCP.

Easy installation and commissioning using a smartphone app (available for iOS and Android).

The Egatel Gen3 terminal has also been integrated into a KNS A6-MK2 maritime antenna (see Figure 8), with good results in terms of quality of service. This opens the way to new IoT-type of services on small boats, with a terminal cost much lower than typical two-way terminals.

### 4.2. Digital Phased Array Antennas and Compact Terminals

The spread spectrum nature of F-SIM also makes it an interesting choice for being used in conjunction with small antennas. The use of small antennas has, in general, a strong impact on the system performance:In the forward link, the received SNR degrades both for the gain loss due to the reduced antenna size and for the increased interference received from adjacent satellites in the GEO arc. Therefore, negative SNR values are typically used, with modulation such as DVB-S2X very low signal-to-noise ratio (down to −9 dB) modes (VL-SNR);In the return link, the limiting factor becomes the aggregate power flux density (PFD) limitation towards adjacent satellites, which translates into a low aggregate achievable spectral efficiency and requires a good terminal pointing. This throughput efficiency still makes the solution attractive compared to the current commercial offer from satellite mobile operators.

The large spreading factors supported by F-SIM reduce the emitted power density from each terminal under the thermal noise level, making it easier to comply with emission masks. As a matter of fact, a badly pointed terminal (e.g., due to a fast mobile platform movement) will generate a very low PFD to the adjacent position, and it can be assumed that the aggregate PFD is well centered around the satellite orbital location actually used.

Another interesting factor is that the same band can be shared with large (fixed) terminals, both in the forward link using DVB-S2X adaptive coding and modulation and in the return link using different spreading factors—therefore, the global system efficiency is only marginally reduced.

Small antennas with electronic steering had a high cost in the past, but recent developments open the way to full terminals well under the USD 1000 threshold, which can target use cases with low volume requirements where the higher transmission cost (due to reduced efficiency) is justified. Some experiments are ongoing with flat panel manufacturers, in order to develop solutions both for fixed use (20 × 20 cm antenna easy to point) or mobile use (flat panel with electronic steering), using different technologies.

Satixfy has developed an integrated antenna/modem that supports F-SIM, including the DVB-S2X VL-SNR modes required in the forward link for such a small antenna. The “Diamond” Ku-band antenna shown in Figure 9 has 256 elements with electronic steering, a nominal EIRP of 32 dBW and a G/T of 2 dB/K at boresight and embeds GPS, Wi-Fi and Bluetooth. Its size is 30 × 35 cm, and its power consumption ranges from 0.5 (sleep) to 60 (receive only) to 90 W (transmit mode).

The antenna works in the time-division multiplex mode, i.e., it cannot receive data while transmitting. Work is ongoing to extend the DVB-S2 protocol by adopting some time slicing techniques—this will, at the same time, reduce the average power consumption to a few watts, according to the use case, and prevent any potential transmit/receive packet collisions.

The integrated antenna/modem is therefore an appropriate solution for (a) fixed installations where no space is available, or ease of installation is required; and (b) mobility environments such as trucks and small boats.

### 4.3. The HyperCube Platform (MBI)

The first F-SIM gateway commercial solution, named HyperCube, was developed by MBI in 2014 in collaboration with Eutelsat and the support of the European Space Agency (ESA) for the operation of the Eutelsat network in combination with SmartLNB terminals. The HyperCube platform is one of several different hardware and protocol evolutions of the first software-defined radio (SDR)-based prototype gateway used in 2009 to receive the very first E-SSA transmission (S-MIM) over the Eutelsat W2A satellite operating in the S band.

The HyperCube platform is a bidirectional, satellite-based interactive system that provides an IP-based, fully compatible and transparent communication between a population of SmartLNBs (the satellite terminals) and the Internet backbone. It has been developed and commercially provided by MBI. 

A high-level architecture of the HyperCube platform is illustrated in Figure 10, in which the hub controls both the forward link (FL) gateway (GW) and the return link (RL) GW.

As it can be seen in Figure 10, HyperCube supports an IP-routed environment, in which the hub behaves as an IP router across the different network segments. Customer-based specific routing policies are also supported. The terminal at the remote side can behave as an IP router too, which is capable managing its own local area network (LAN). A more detailed HyperCube platform functional block diagram is shown in Figure 11. A compact and turn-key HyperCube platform in its entry level version can easily fit within a 19″ rack, as shown in Figure 12.

The HyperCube platform elements shown in Figure 11 have been integrated in a single 19″ rack hosted in the satellite GW (see Figure 12). In the case of a multi-beam high-throughput satellite (HTS) (e.g., Eutelsat Ka-Sat), each GW serves a subset of the HTS beams. The size of the beam cluster served by each gateway is related to the feeder link bandwidth. In this case, a number of FL/RL HyperCube GWs have to be geographically distributed. Such remote FL/RL GWs have been connected to the central hub through an IP-based backbone thanks to the modular HyperCube architecture.

The RL side of the GW is based on a software-defined radio (SDR) architecture (see Figure 13): it adopts an off-the-shelf SDR device to perform down-conversion to the baseband, and to convert, through analogue-to-digital, the aggregated signal. Then, the I–Q digital samples are delivered through a 10GBE connection to the demodulation chain composed of a number of demodulation nodes connected in a cascade configuration. Each node is built using a rack-mountable server equipped with a central processing unit (CPU) and graphical processor unit (GPU) that share the computational processing. All the digital signal processing (DSP) algorithms required to demodulate F-SIM bursts are executed in the software by running the C++ object code hosted on each demodulation node. DSP algorithms are based on the Intel Math Kernel Library and the NVIDIA Compute Unified Device Architecture (CUDA), respectively, for CPUs and GPUs.

This SDR approach has changed the way gateway demodulator processing is developed, once only based on the traditional field-programmable gate array (FPGA) approach. In particular, this approach allows increasing the performance by adopting the newest CPUs/GPUs released on the market by major manufacturers, with marginal costs for the porting of the code. After about ten years of continuous operations, the viability of the SDR approach for the ground segment of the random access E-SSA system has reached full maturity.

One of the main advantages of the adopted software approach is the simplicity of the porting of the software on different hardware platforms and the ability to easily profit from the advances made by new multipurpose GPUs continuously improved to serve larger markets (e.g., artificial intelligence (AI), 3D gaming, blockchain processing, video encoding and other computationally intensive applications). This design approach ensures that the latest state-of-the-art COTS equipment can be quickly adopted for the commercial demodulator. At the same time, the SDR-based design makes very low-cost E-SSA testbed PC-based implementations available. Such compact testbed including few user terminals, faithful traffic emulation, a gateway demodulator and an ancillary monitor and control software has been used to perform early live satellite demonstrations of S-MIM and F-SIM IoT solutions, to assess their performance in a laboratory environment and to assist terminal manufacturers.

The SDR gateway comes with a high degree of scalability. In fact, an additional node(s) can be added in a cascade configuration using 10GBE links as the system throughput requirement grows. Thanks to this, the entry-level solution is very light, compact and cost-effective as additional demodulation nodes can be added later on. The random access protocol used by F-SIM is based on the innovative iterative detection process detailed in Section 3. This type of signal processing requires a significant computational burden at the gateway. The following tasks are independently performed for every received packet by each demodulation node present within the demodulation chain: preamble detection, channel estimation, demodulation and decoding of the control and data channels, regeneration of a baseband replica of the received packet and, finally, cancellation from the sliding window memory of the decoded packets using the regenerated replicas. The number of packets that can be demodulated and cancelled from the sliding window memory depends on the HW resources of each node. Once a node has performed the maximum number of SIC it is able to perform, the window sample memory (where the demodulated packets have been cancelled) is passed, for further processing, to the following demodulation node, etc.

### 4.4. Current Deployment Status

Eutelsat recently launched the IoT First service, dedicated to the IoT market, based on the HyperCube hub and SmartLNB terminals. The service, which is operated in the Ku band, is available in different regions of the world such as:Europe, on Eutelsat 10A (see Figure 14);Africa sub-Sahara, on Eutelsat 7B;United States and Mexico (CONUS coverage), on Eutelsat 117WA (see Figure 15);North and South America (pan-America coverage), on Eutelsat 117WA.

Moreover, trials are ongoing in the Far East at the time of writing.

The terminals are now at the third hardware generation. The latest generation terminals are smaller, lighter and overall cheaper than previous ones. Their architecture is based on the ST Cardiff3 chipset [21], which has been customized to both receive the DVB-S2 signal and to transmit the F-SIM waveform.

## 5. Ongoing R&D

In recent years, a number of adaptations and enhancements to the existing specifications have been proposed in order to maximize the spectral efficiency of the RA E-SSA-based scheme, and to adapt it to different utilization scenarios. In particular, some of the current R&D activities are related to the possibility of operating a massive number of low-cost IoT terminals also using non-geostationary satellite orbit (NGSO) satellites and smaller channelization. The main examples of the current R&D tracks are illustrated in the following sub-sections.

### 5.1. The Massive Project

The ESA funded the ARTES AT MASSIVE project involving MBI S.r.L. and AIRBUS Italia S.p.A. (ADSR), aiming at improving the spectral efficiency of the E-SSA random access scheme by employing a linear minimum mean square error (MMSE) detector rather than a conventional single-user matched filter (SUMF) at the receiver. The above approach is justified by the fact that BPSK modulation is optimal when an SUMF detector is used and provides robustness to carrier phase noise. The multiple access spectral efficiency is maximized by single-user coding and decoding, and by implementing the SIC process described above. In this way, the complexity increases linearly with the number of users.

To further boost the spectral efficiency performance of the spread spectrum random access scheme, especially in the case of a reduced power unbalance, a linear MMSE (LMMSE) detector can be used prior the SIC process [22]. In this case, BPSK modulation is not optimal, while QPSK modulation shows asymptotic optimality [22]. The adoption of QPSK modulation requires a modification of the waveform design. In particular, the quadrature multiplexing between PCCH and PDCH is replaced by time domain multiplexing (TDM) between the two channels. In the following, such E-SSA access scheme employing MMSE will be referred to as ME-SSA for short. Starting from the analysis reported in [23], the MASSIVE project implemented the LMMSE detector in the form of a multistage despreader (MSD) [24,25]. The latter actually approximates an LMMSE detector, with the accuracy of approximation improving with the number of stages. The LMMSE detector requires, in fact, the inversion of the covariance matrix R, which is prohibitive in a real-time scenario. The MSD approximates the inverse of the covariance matrix by a polynomial expansion in R, that is: $R^{-1}≃\overset{N}{\underset{n=1}{\sum }}w_{n}R^{n}$. A number of stages N equal to 2 or 3 is shown to provide a good approximation for the most typical scenarios. The weight $w_{n}$ is properly chosen to approximate the LMMSE detector, i.e., $\overset{N}{\underset{n=1}{\sum }}w_{n}R^{n}≃{\left(R+N_{0}I\right)}^{-1}$, $N_{0}$ being the noise power.

Assuming K bursts to be demodulated, the MSD implementation consists of a sequence of N identical stages, as depicted in Figure 16. First, the K received noisy bursts are individually despread. Then, they are input to the first stage, where the symbols are respread, time and frequency offset are restored and the so-obtained bursts are summed together. The resulting signal is sent in parallel to K lines, one per burst, where the relevant burst time and frequency offsets are corrected, and then despreading is applied. The symbols obtained in this way for each burst are input to the following stage, which performs the same processing as the previous one, until all the stages have been processed. The output symbols of each stage are also weighted and summed together. Once the final stage has been executed, such weighted sum represents the K despread bursts as they were obtained by a linear MMSE detector. Different approaches are available in the literature for weight computation. In this project, the approach proposed in [24] was adopted, as it provides a very good approximation of the LMMSE, accounting for the power each symbol is received with and the pulse shaping used, other than the noise power and the system load, i.e., the ratio between the number of interfering bursts and the spreading factor. In conclusion, ME-SSA should provide an improved spectral efficiency (theoretically, up to 50%), especially with low SFs/a high bit rate where E-SSA performances are degraded, without increasing the complexity at the user terminal side. The complexity increase is at the gateway side, where, instead of performing one despreading per burst as in the SUMF case, spreading and despreading operations are carried out for each burst at each stage of the MSD.

Figure 17 shows the throughput and packet loss ratio (PLR) vs. offered MAC load comparison between two E-SSA and ME-SSA waveforms, both with the same FEC coding rate (3GPP Turbo code with rate 1/3), a payload length of 1200 bits, a spreading factor of 16 and a chip rate of $R_{c}=1.92$ Mcps, and for two power randomization (PR) cases: a power randomization uniformly distributed between 0 and 5 dB; no power randomization. In practice, the ME-SSA waveform corresponds to the adaptation of the F-SIM one. The minimum received E_s_/N_0_ is 6 dB for both waveforms. The only difference, as explained above, is that the E-SSA waveform adopts BPSK modulation and the SUMF receiver, whereas the ME-SSA one adopts QPSK modulation and the MSD receiver. Simulation results were obtained assuming six successive interference cancellation (SIC) loops and an actual packet demodulator with frequency and phase estimators enabled, but with an ideal preamble searcher. This means that all the burst timestamps at the receiver are known. As expected, in such a low-SF regime, the ME-SSA waveform outperforms the E-SSA one, improving the overall throughput. Without PR, at PER $≃10^{-3}$, the throughput improvement obtained with ME-SSA is about 50%, with 5 dB uniform in dB PR, as expected, and such improvement decreases to about 25%.

Figure 18 compares the performance, in terms of throughput vs. SIC loops, obtained by the MSD-based receiver for the same ME-SSA waveform introduced above, for two offered load cases and for two preamble searcher (PS) cases: the ideal PS and the actual PS. From Figure 18, it can be seen that, to obtain the same performance of the ideal PS case for the two offered load cases taken into account, more SIC loops are required as not all the interfering bursts are immediately known. Additionally, the maximum throughput obtained with a real PS was about 1 bit/chip, corresponding to a performance loss of about 15% compared to the maximum throughput obtained with ideal PS case, depicted in Figure 18. In summary, the actual receiver parameters (PS threshold, buffer size and dispatch, etc.) shall be carefully tuned to gain a trade-off between performance and latency when the MSD is employed.

### 5.2. The GEMMA Project

The ESA ARTES AT GEMMA project (MBI S.r.L. and AIRBUS Italia S.p.A. (ADSR)). S-MIM and F-SIM rely on DVB-SH and DVB-S2 standards, respectively, on the forward link (FL). Fulfilling the need for an adaptation of the FL air specifications to also operate mobile IoT terminals in different scenarios is the goal of this project. In particular, the main requirements are as follows:(a)The support of both GEO and LEO scenarios, keeping the user terminal inexpensive and easy to operate;(b)The support of both fixed and mobile terminal applications;(c)The support of different types of applications (point-to-point, multicast and broadcast) and data rates also enabling those services relying on data transfer to terminals (e.g., firmware upgrade);(d)The possibility to implement different Tx/Rx activity modes that could help in reducing the power consumption of the terminal; the possibility to implement loop functionalities (e.g., ARQ, congestion control, power randomization and/or variable ModCods/SFs) for network management to increase the system capacity.

The new air interface was designed capitalizing on the most suitable technology solutions adopted in satellite standards such as DVB-S2, DVB-SH and ETSI-SDR. The channel coding is based on the 3GPP LTE Turbo codes as they provide a good trade-off between performance and complexity. They perform well at low coding rates compared to LDPC codes which perform better at higher coding rates but have a higher memory requirement.

A channel-programmable length time interleaver is employed to counteract outages due to shadowing or short blockages in mobile scenarios. It is based on convolutional interleaving, as used in DVB-SH, since, compared to block interleavers, it provides a reduction in the memory occupation by a factor 2.

The transmission is organized in equal length frames with a constant pilot symbol spacing to ease the acquisition process at the terminal. In order to support different quality of services (Q.o.S.), several physical layer pipes (PLP) are defined, each of them identified by a combination of physical layer parameters (modulation order, coding rate, convolutional interleaver parameters) and mapped into a frame.

A spreading up to factor 4, common for all the frames, can be applied in order to improve the minimum SNR demodulation threshold. This can be useful, for instance, in a LEO scenario where, towards the poles, a terminal may see more than one LEO satellite belonging to the same constellation. In this case, the second satellite in view creates co-channel interference, decreasing the received signal over noise plus interference ratio. Another case could be the one where lower gain antennas are employed at the terminal so that such gain loss is compensated by the processing gain.

The synergy between MASSIVE and GEMMA projects focuses on different performance requirements and scenarios. The main research activity is aimed at adapting the RA E-SSA protocol to the LEO constellation with a smaller available bandwidth (on the order of hundreds of kHz). Within this context, the main challenges consist in dealing with a higher Doppler shift range and a non-negligible Doppler rate typical of LEO scenarios. This has an impact on the PS, which shall be robust to these effects, and calls for the need for a Doppler rate estimator at the receiver. Furthermore, the LEO scenario requires low-power consumption demodulation algorithms, as demodulation takes place on board and is typically based on low-cost COTS hardware.

The synergy between these two projects also led to the definition of an air interface to be adopted in GEO Ku and Ka band scenarios, with terminals equipped with low-gain flat (e.g., patch array) antennas. Such air interface, named IURA (IoT Universal Radio Access), is based on E-SSA on the return link, which is able to work at very low carrier-to-noise power ratios (i.e., C/N below −20 dB) thanks to the processing gain provided by the large spreading values (up to 256), and an evolved GEMMA waveform on the forward link, in order to deal with values of C/N below −15 dB. These extended C/N ranges enable the possibility to equip the terminals with small low-gain antennas, e.g., an 8 × 8 patch array, and still be able to operate in typical GEO Ku and Ka band scenarios. This, combined with the offered data rates ranging from some kilobits per second to some tens of kilobits per second with larger flat antennas and the possibility to manage a very high number of devices which sporadically transmit few data bursts, makes it suitable to develop a simple and low-cost terminal for IoT and medium-data rate scenarios.

We report the following laboratory results for Use Case 2 (UC#2) corresponding to a very small antenna and very low power consumption terminals, with antenna sizes in the orders of few centimeters (e.g., 6 × 6 cm), transmitting over the GEO Ku band. The low EIRP and power consumption calls for a waveform with a high SF and a low power spread. A scenario with terminals transmitting an IURA waveform with an SF = 256 and a chip rate of $R_{c}=220$ kchip/s, with a minimum received C/N = −23 dB and limited power randomization (3 dB), is representative of this use case. For this scenario, the following remarks are in order:

(i) Thanks to the E-SSA multiple access, a certain number of simultaneous transmitting terminals (STT), each with the same EIRP, access the same spectrum resource (a band *B*) at the same time. The EIRP(θ) towards a direction which is θ degrees off-axis w.r.t. to the maximum radiation direction depends on the antenna radiation pattern. Hence, the overall off-axis EIRP density at θ degrees is given by EIRP(θ) − 10log_10_(*B*) + 10log_10_(STT).

(ii) A number of international regulations such as ITU, FCC and ETSI define the maximum level of the off-axis emission density (OAED) that can be emitted using Ku FSS bands by all terrestrial terminals operated within a given channel.

(iii) Due to the adoption of small antennas such as a patch array of 64 elements, the antenna radiation pattern is denoted by a very wide main lobe and high side lobes, increasing the EIRP density in the off-axis direction compared to the case where a bigger antenna is used.

To this end, Figure 19 compares the OAED mask (dashed line) provided by the ITU-R recommendation [26] and the off-axis EIRP density (red line) that would be obtained by an IURA terminal in UC#2, that is, considering an EIRP = −13 dBW and a 64-element patch array antenna with a maximum gain of about 21 dB. The minimum distance between the latter and the OAED mask denotes, on the logarithmic scale, the maximum STT (STT_max_) that can be operated before violating the constraint. Such value turns out to be STT_max_ = 15, and the aggregated EIRP density is represented by the yellow curve. Table 2 summarizes the relevant performance of the IURA waveform. It is worth to point out how the E-SSA protocol makes it possible the use of such very small antennas. Indeed, the latter call for a high spreading factor in order to compensate the low EIRP with the processing gain at the receiver, and thus to successfully close the link budget. This would not be possible with non-spread ALOHA-based protocols, where, in addition, the collisions would further degrade the performance without being able to provide 15 STTs. Furthermore, orthogonal access schemes, such as time-division multiple access, would yield a waste of band and imply a higher signaling. Hence, the E-SSA protocol stands as an effective and suitable solution also for this use case.

### 5.3. The IoT-SATBACK and 5G-SENSOR@SEA Projects

An example of studies of upper layer enhancements and optimizations is represented by the ESA project IOT-SATBACK [27] (MBI S.r.L. Pisa, Italy and Software Radio Systems Ltd., Cork, UK), whose primary objective is to design, develop, test and demonstrate a testbed capable of providing satellite backhauling services for future NB-IoT. The targeted improvement is to enable new satellite communication services for backhauling M2M and Internet of Things communications, and this is achieved by means of a component named the IP Optimizer which implements optimizations of F-SIM layer 2 and upper layers. The optimization includes techniques for the reduction in the overheads and the IP payload compression which contributes to increasing the backhauling spectral efficiency.

The outcomes of the IOT SATBACK studies have a natural follow-up in the ARTES C&G project named 5G SENSOR@SEA, where a complete end-to-end system including a satellite part based on enhancements of the optimizations mentioned above is going to be tested in a real operational scenario to transmit sensor data from cargo ships in open or near sea to an IoT platform. The final target is a complete platform which can be sold as a commercial product to be provided to cargo ship companies and other companies interested in container, goods or fleet monitoring in the logistic and transport area.

### 5.4. Putting an IoT Gateway in Space

Differently from geostationary satellites, for LEO IoT satellite constellations, it is not always possible to ensure continuous ground-based connectivity by using a limited number of GWs. In this case, the uplinked packet demodulation shall take place on board the satellite in order to reduce the requirements on the feeder link. Decoded packets shall then be stored on board and dumped to ground via the feeder downlink when the ground station is in view, or using inter-satellite link (ISL)-based connectivity if this is available.

According to public information, this demodulate-and-store approach is adopted by various LEO systems and, in particular, using the E-SSA protocol by the Dutch company Hiber on its CubeSats [28,29].

#### 5.4.1. On-Board NOMA Demodulator Implementation

The use of GPU-based solutions for the on-board digital signal processing represents the next frontier in the development of cost-effective small satellites easily adaptable to different scenarios thanks to the flexibility of a fully software-based payload. Key features of the proposed low-power on-board GPU for real-time demodulation are as follows:Combining advanced access, modulation and coding techniques with a fully programmable SDR/GPU architecture;Allowing multiple applications to be tested and validated and/or a continuous upgrade and optimization of on-board performance;Reducing the obsolescence of on-board processing satellite infrastructures thanks to the possibility to upgrade the firmware;Leveraging upon high-performance VLSI chipset widely used for artificial intelligence (AI), 3D gaming, blockchain processing, video encoding and other computationally intensive applications. The processing power of those chipsets is rapidly growing and de facto promises to sustain the future operational system performance;Leveraging upon the first on-ground advanced communication system which uses a full SDR/GPU-based gateway system already deployed in four continents.

The on-board CPU/GPU platform can also be used to operate different return and forward link air interfaces, thus maximizing the flexibility and scalability of the solution. Figure 20 shows how the on-board CPU/GPU module can be integrated with the other components of a small satellite. The codebase for E-SSA signal demodulation is developed based on an inherited code from the available E-SSA demodulator, running on similar NVIDIA cards.

Optimization efforts are needed to maximize the performance of the demodulator:Adapting and optimizing the telecommunication performances by maximizing the achievable throughput while, at the same time, minimizing the DC consumption;Rewriting part of the software to better cope with the flying environment where radiation occasionally causes software errors (i.e., bitflips).

In relation to the latter point, any satellite on-board embedded electronic system is expected to carry out its mission despite faults that may occur due to, e.g., high-energy particles and cosmic radiation. Fault tolerance to such events can be created by design at two levels [30]. Firstly, this can be achieved at the hardware level by means of parallel processing units (PPUs) that run the same micro-code and are provided with the same input. Whenever their output differs, this means that a failure has occurred. Recovery is conducted in the hardware by means of a voting system (with three or more parallel units, the correct result is assumed to be the one given by the majority) or roll-back (the last batch of instructions is executed again, until the same result is achieved from all). Secondly, at the software level, an error detection and correction (EDAC) approach should be implemented. This can be conducted by storing the checksum of all routines and static data structures and periodically checking that there have been no bitflips.

The design, development and validation of such a complex embedded system implementing such protections is quite high, and rarely justified. In this case, re-use of COTS hardware or software is problematic because these protection measures are not implemented in typical commercial products (with very few exceptions, e.g., micro-controllers for high-power electric systems). Furthermore, it is not possible to benefit from mature, stable and proven software stacks coming from the open-source community [31]. In particular, in [32], the authors highlight the advantages of using open-source software, especially the Linux Operating System, on COTS hardware, as a means to achieve short development cycles and, hence, foster innovation in the space segment.

A possible way to add redundancy at the sub-system level without re-designing the whole demodulator is the following. Firstly, the E-SSA software is enriched by self-monitoring functions to detect memory corruptions. Such functions may combine checksum verification with a keep-alive mechanism that is triggered periodically by the OS. Secondly, multiple redundant boards, equipped with the same software and fed by the same input, can be installed as the actual on-board payload. The boards will either manage one another to guarantee that at least one is performing correctly, as proposed in [33], or rely on an external controller that is able to reboot a board in some conditions. Since the NVIDIA Jetson is equipped with GPIOs (general purpose inputs/outputs), the controller could be a simple hardware watchdog circuit with relays (clearly, the controller itself must be defined to be fault-tolerant, but this task is largely affordable since its functions are extremely simple).

The baseline approach has been that of an on-board processing based on the NVIDIA Jetson TX2 board or the latest NVIDIA Jetson AGX Xavier [33] board (see Figure 21) that has already been studied for CubeSat applications [34]. In particular, the NVIDIA Jetson AGX Xavier is an artificial intelligence (AI) computer for autonomous machines, delivering GPU workstation performance with an unparalleled 32 TeraOPS (TOPS) of peak computing in a compact 100 × 87 mm module form factor with user-configurable operating modes at 10 W, 15 W and 30 W (the power consumption can be reduced to 7.5 W with lower performance or reducing functionalities).

The main demodulation results based on recent laboratory tests on the Jetson TX2 and Xavier in a non-optimized environment are listed in Table 3.

#### 5.4.2. Laboratory Test Results

This section shows laboratory test results for the IURA use case (UC#3) which is representative of a regenerative on-board processing (OBP) scenario, typical for LEO regenerative satellites. The terminal is battery- and low-powered, and the on-board processing resources are very limited compared to the on-ground demodulation. Hence, a small channelization, small spreading factor values and few SIC iterations shall be considered for this case. Figure 22 and Figure 23 depict the throughput and PLR performance, respectively, considering terminals transmitting an IURA waveform with a chip rate of $R_{c}=220$ kchip/s, an SF = 16, a minimum C/N = −15 dB and 6 dB of available power randomization. Such range, for instance, can be due to the path loss difference between two terminals, one that observes the satellite with the minimum elevation angle (e.g., at about 30 degrees) and another one with the maximum elevation angle (i.e., 90 degrees). The throughout and PLR performance are shown for different numbers of SIC iterations, specifically, from one SIC to four SIC iterations. The result is that, depending on the OBP capabilities, the performance can vary from about 0.1 (one SIC iteration) up to about 0.55 bit/chip (four SIC iterations), at PLR = 10^−2^. A summary of the results is shown in Table 4. For the sake of clearness, the generic SIC iteration consists of the sequence of the following processing blocks: acquisition, demodulation, regeneration and cancellation. Hence, the demodulation performance with zero SIC iterations does not benefit from cancellation, which is carried out only at the end. Additionally, it is worth to stress the reason why the throughput performance of UC#3 is lower than UC#1: UC#3 results were obtained with a much less powerful hardware, as the UC#3 demodulator is on board the satellite, and hence shall have low power consumption with a reduced number of SICs. Conversely, on-ground receivers in UC#1 may employ a much more powerful demodulator hardware and a higher number of SICs, thus providing a superior throughput performance.

## 6. Possible Commonalities with 5G mMTC

At the beginning of the first 5G activities within the 3GPP standardization group (i.e., Release 15), advanced NOMA techniques were proposed and investigated for massive machine-type communication (mMTC) services. A very good summary and categorization of the proposed NOMA schemes are reported in [35,36], while a comparative performance analysis based on selected 5G system-level assumptions has been summarized in [37]. Even though none of those NOMA candidates have been selected and standardized thus far for 5G mMTC services, the different proposals from the terrestrial wireless industry players can be grouped into three main multiple access (MA) categories: (a) codebook-based; (b) sequence-based; (c) interleaver/scrambler-based. Hereafter, a very brief summary of the key features is presented in order to understand the possible commonalities with the S-MIM/F-SIM technology.

Codebook-based MA maps the user data packet stream in a multi-dimensional codeword belonging to a codebook. The mapping is conducted in a way to achieve signal spreading and introduce zero elements to mitigate inter-user interference. The decoding process is obtained through a relatively complex iterative message passing algorithm.Sequence-based MA exploits non-orthogonal complex number sequences (short or long sequences) to separate users sharing the same spectrum, thus easing the multi-user detection process. Affordable complexity linear MMSE plus SIC or parallel interference cancellation (PIC) is proposed for the packet detection.Finally, interleaver/scrambler-based MA utilizes different interleavers to separate users sharing the same bandwidth. Some repetition/scrambling is also adopted to spread the signals and achieve some interference-averaging effect. Depending on the size of the interleaved bit stream, simpler MMSE-SIC or more complex soft SIC decoding techniques will be used.

From the proposed NOMA categorization in 3GPP, it shall be easier to associate the E-SSA random access protocol at the sequence-based MA techniques, and, in particular, specific commonalities can be found in the MUSA [37] and in the RSMA [38] proposals. As already said, the current 5G standard (from Release 15 to Release 17) has not included NOMA techniques for mMTC services. Nevertheless, the advanced RA techniques (such as E-SSA or others) may benefit in the future to be part of 5G and beyond terrestrial standards by exploiting the current promising trend to work for a full and seamless integration between satellite and terrestrial networks.

The rapid evolution of NGSO systems, resulting today in the deployment of thousands of LEO satellites, and the renewed interest in 3GPP in the integration between satellite-based and terrestrial-based 5G systems are good precursors to the integration of NOMA-based multiple access systems, starting from the 3GPP Release 18. A proposal inspired by the long development and operational experience cumulated in the last ten years, in the actual implementation of E-SSA-based systems, may pave the way to its consideration for future mMTC satellite and terrestrial applications. NOMA-based technologies, such as E-SSA, have demonstrated the ability to operate in mobility and in GSO and non-GSO-based orbits and provide a massive scalability, high efficiency and low user terminal cost.

## 7. Conclusions

NOMA technologies were pioneered about ten years ago in the satellite domain to exploit, at best, the growing demand for IoT applications, characterized by very large populations of users, sporadically transmitting small to medium-size packets with low-cost, easy to install terminals.

The key features of the developed E-SSA NOMA system are as follows:The achievable very high spectral efficiency while operating in pure random access mode. Networks today in operation in some operational configurations are reaching close to 2 bits/s/Hz efficiency;The easy network scalability and the support in the same band of multiple configurations to match the different application needs;The very low-cost and low-power two-way satellite terminals developed and industrialized crossing, for the first time, the USD 100 cost threshold, pushing the satellite connectivity market towards the consumer market.

The proposed NOMA system also allows the inclusion, on the same technical platform, of a new class of integrated digital modem and smart antenna systems capable of simplifying the installation and exploitation of satellite-based systems to a level comparable to terrestrial communication systems, i.e., not requiring any manual pointing support.

The development of a common IoT specification at the disposal of competing terminal manufacturers has, for the first time, allowed the development of a satellite-based two-way system able to operate with multiple suppliers and interoperable terminals.

An end-to-end software-defined radio architectural approach has been selected from the very beginning for both the user terminal and, most importantly, the gateway/hub. This new paradigm allows the whole satellite IoT system to evolve in time, both in terms of software upgrades and overall system management.

In recent years, a number of adaptations and enhancements to the existing specifications have been proposed in order to further enhance the spectral efficiency of the E-SSA-based scheme, and to adapt it to different utilization scenarios. In particular, the main requirements addressed are the support of both GEO and LEO scenarios, the possibility of operating a massive number of ultra-low cost IoT terminals and the support of both fixed and mobile IoT applications.

NOMA technology is capable of exploiting, at best, the requirements stemming from IoT applications and, at the same time, adapting well to the specific characteristics of terrestrial and multi-layer satellite networks (GSO and non-GSO). The long development and operational experience cumulated in the last ten years, in the actual implementation of E-SSA-based IoT satellite systems, provides a solid ground for an effective integration of terrestrial and satellite networks for mMTC services in 5G and beyond. The challenge is to combine the NOMA high performance with affordable complexity and an extremely low cost for both the user terminal and the associated infrastructure. The objective is to make mMTC a reliable, affordable and scalable infrastructure for billions of devices to be interconnected.

## Figures and Tables

**Figure 1 sensors-21-04290-f001:**
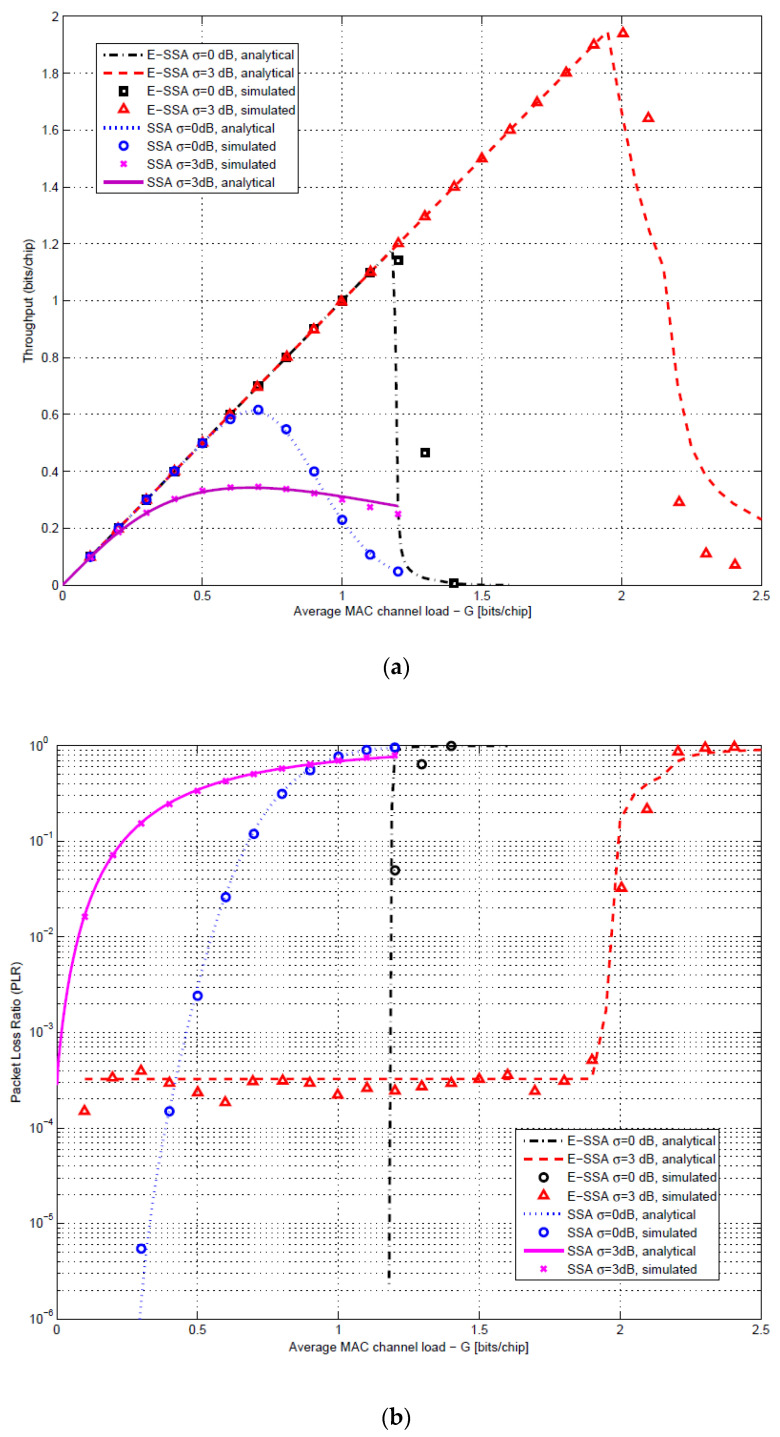
Simulated vs. analytical SSA and E-SSA throughput (**a**) and PLR (**b**) performance with and without lognormal power unbalance from [11], 3GPP forward error correction (FEC) code rate 1/3 with block size of 100 bits, BPSK modulation, spreading factor of 256 and E_s_/N_0_ = 6 dB (© Copyright 2016 John Wiley and Sons).

**Figure 2 sensors-21-04290-f002:**
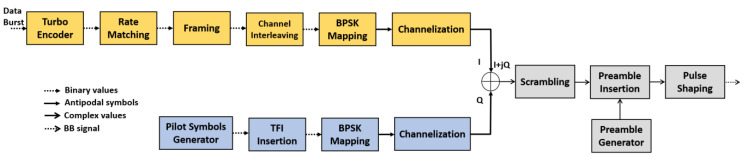
F-SIM transmitter functional block diagram.

**Figure 3 sensors-21-04290-f003:**
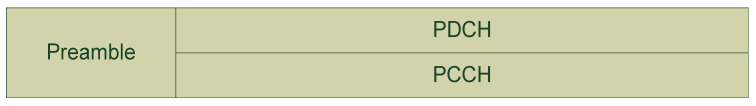
The uplink burst and its constituent parts.

**Figure 4 sensors-21-04290-f004:**
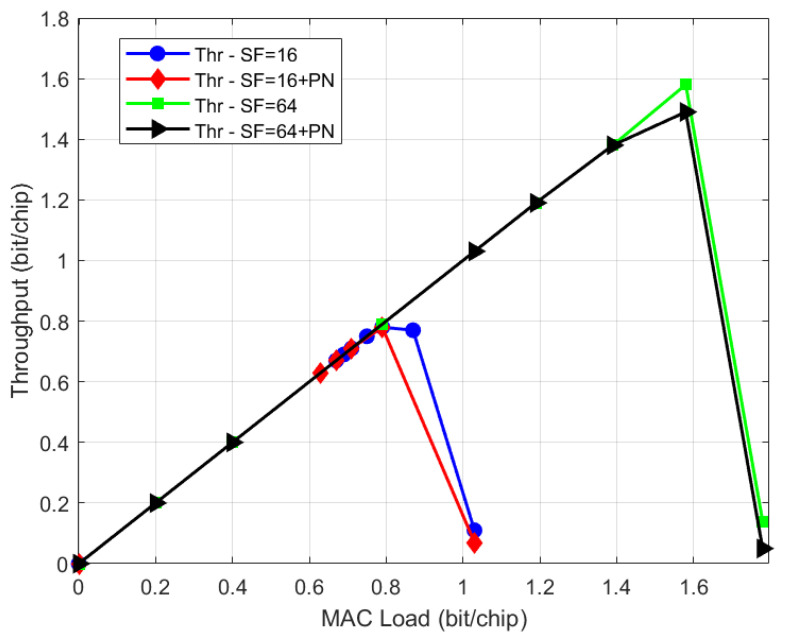
UC#1: F-SIM throughput performance.

**Figure 5 sensors-21-04290-f005:**
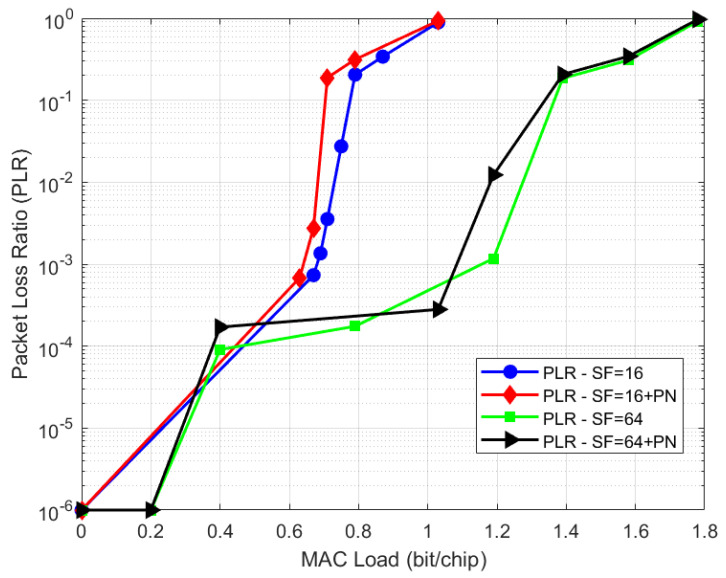
UC#1: F-SIM PLR performance.

**Figure 6 sensors-21-04290-f006:**
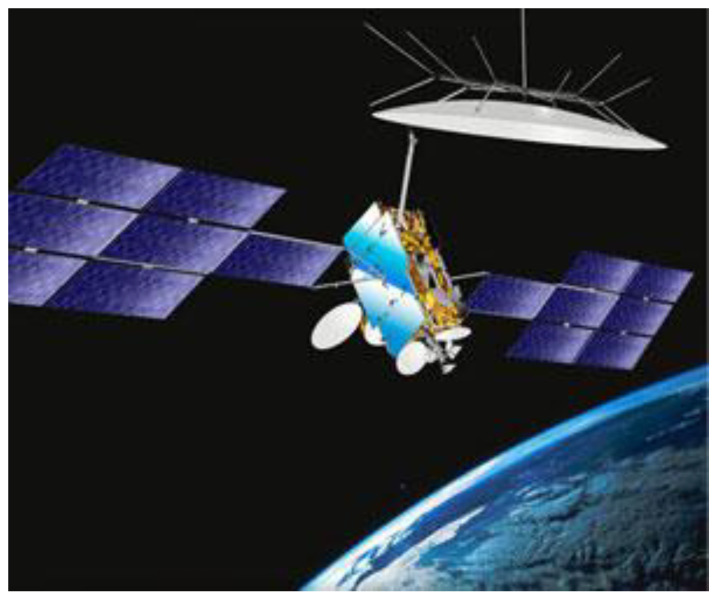
The Eutelsat W2A satellite pictorial image showing the 12 m deployable reflector for the S-band mission.

**Figure 7 sensors-21-04290-f007:**
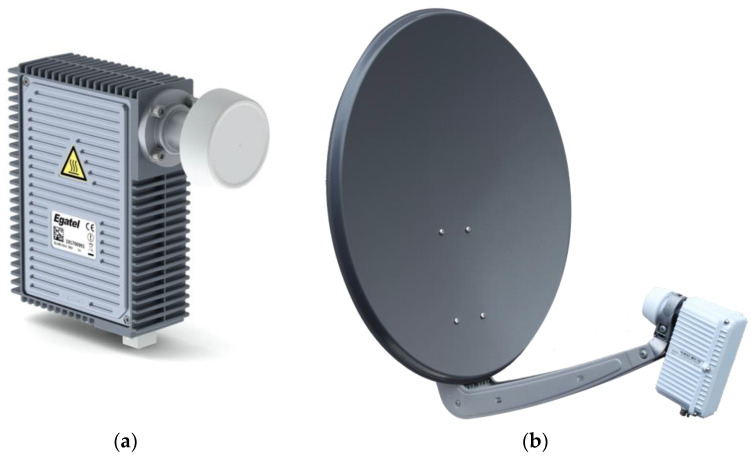
SmartLNB Gen3 terminals from Egatel (**a**) and Ayecka (**b**, mounted on a dish).

**Figure 8 sensors-21-04290-f008:**
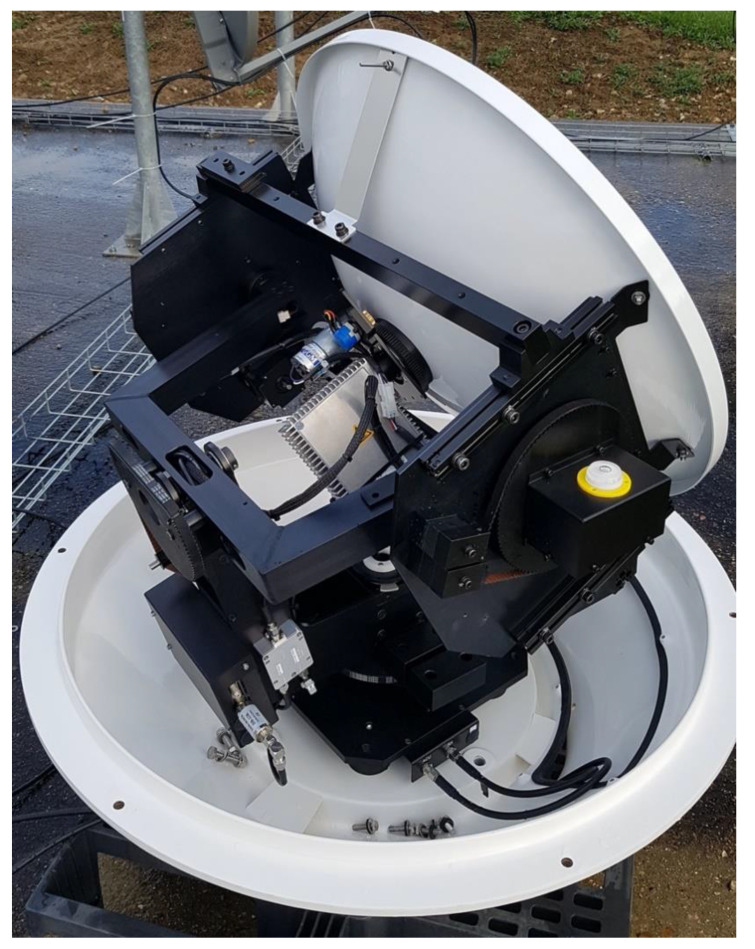
Egatel terminal mounted in a KNS maritime antenna.

**Figure 9 sensors-21-04290-f009:**
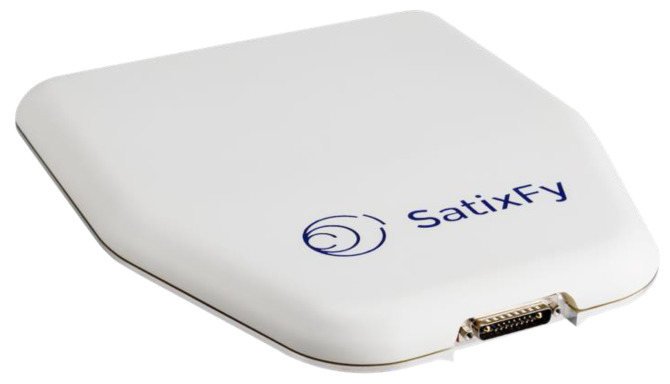
The Satixfy Diamond Ku-band antenna.

**Figure 10 sensors-21-04290-f010:**
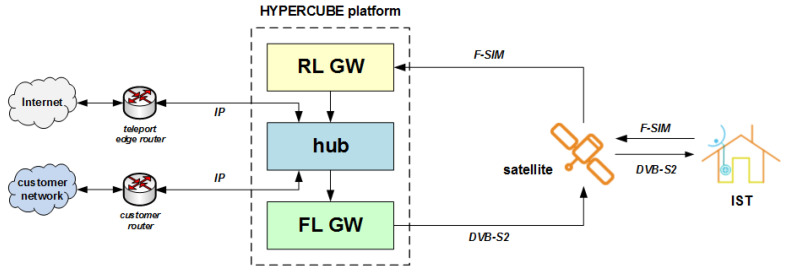
High-level functional block diagram of the HyperCube platform.

**Figure 11 sensors-21-04290-f011:**
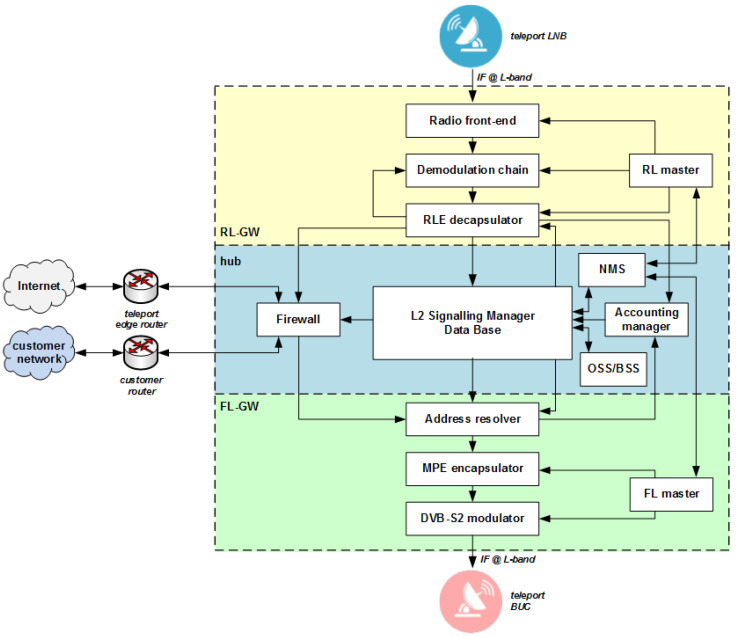
Detailed block diagram of the HyperCube platform.

**Figure 12 sensors-21-04290-f012:**
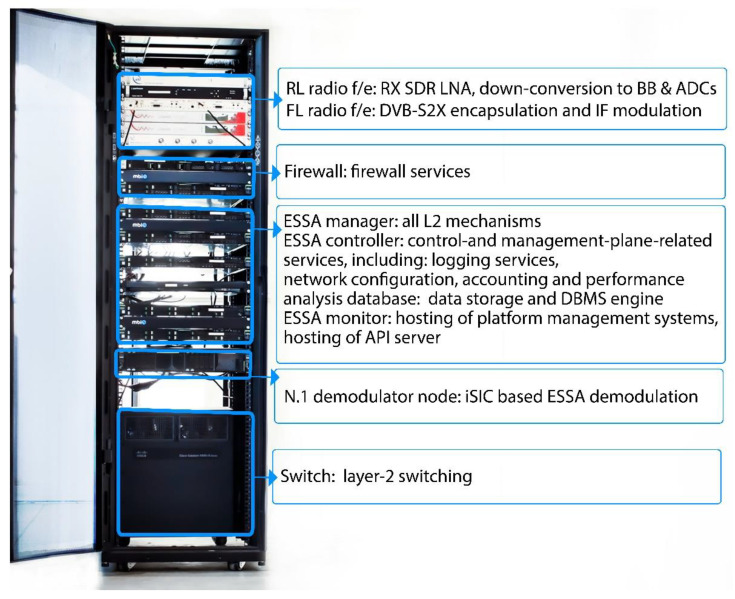
HyperCube platform hosted in a 19″ rack.

**Figure 13 sensors-21-04290-f013:**
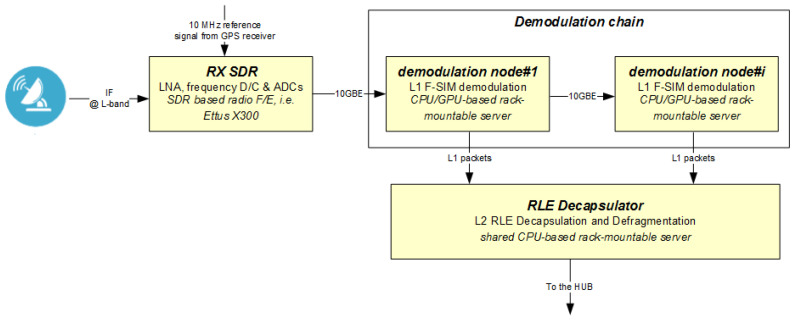
Detailed view of the RL GW.

**Figure 14 sensors-21-04290-f014:**
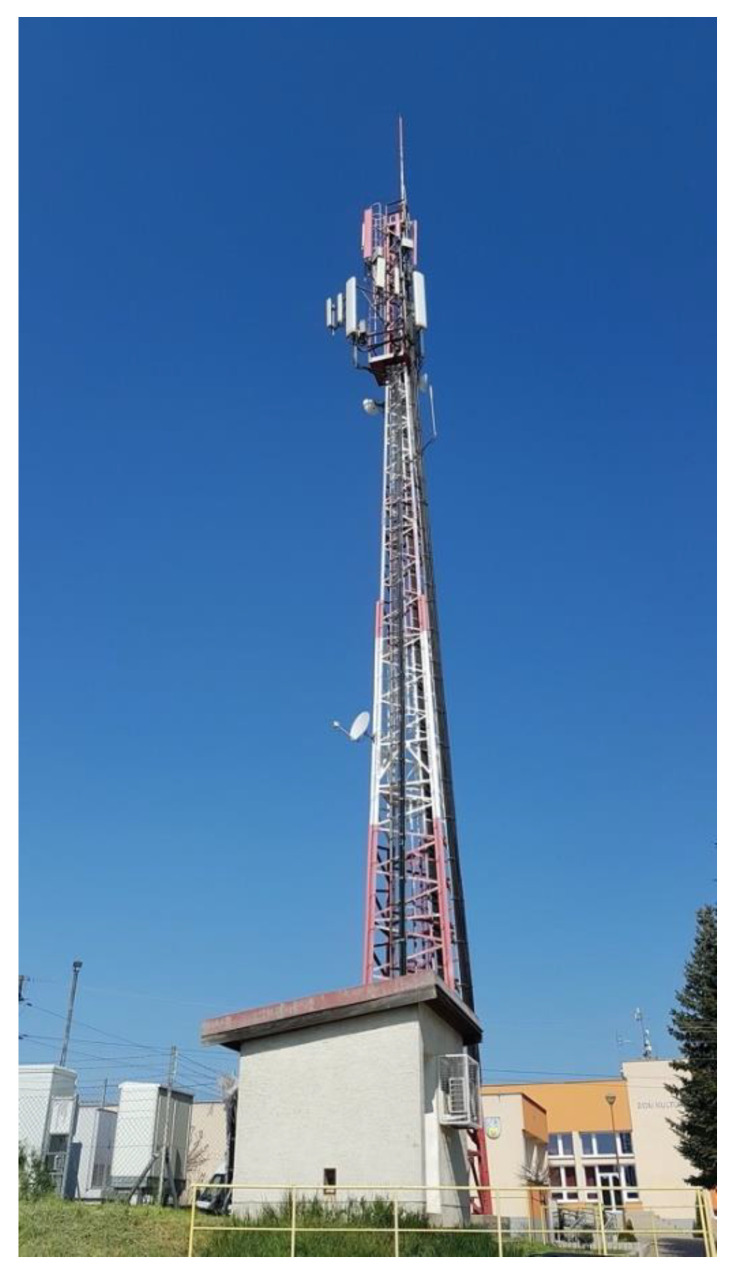
A SmartLNB installed on a communication tower in Slovakia (EUTELSAT 10A coverage).

**Figure 15 sensors-21-04290-f015:**
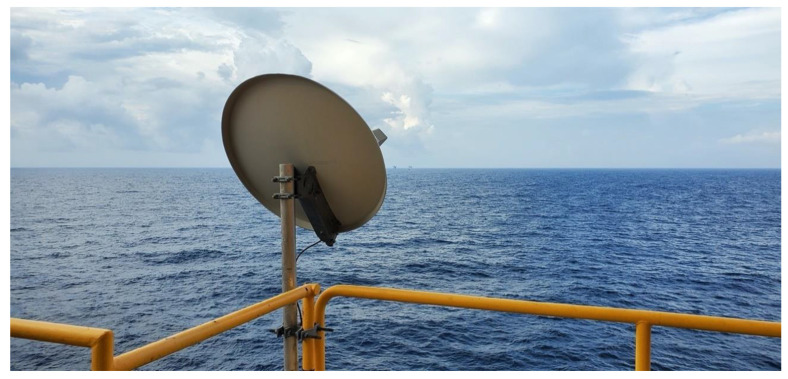
A SmartLNB installed on an oil rig in the Gulf of Mexico (EUTELSAT 117WA coverage).

**Figure 16 sensors-21-04290-f016:**
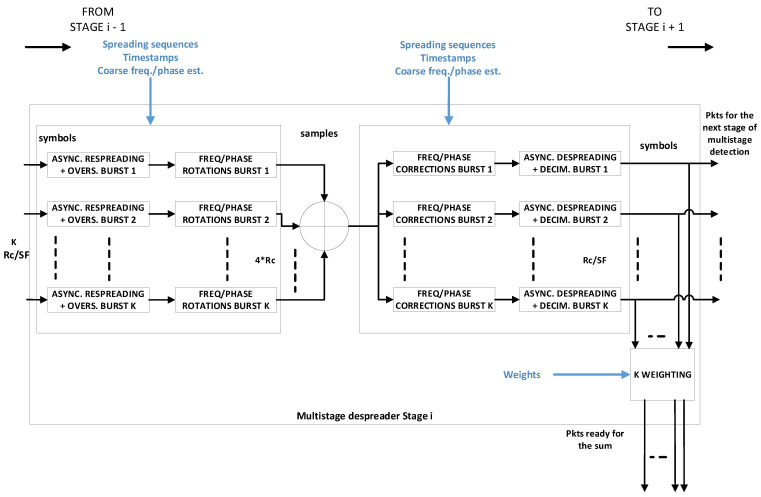
Generic *i*th stage of the multistage despreader.

**Figure 17 sensors-21-04290-f017:**
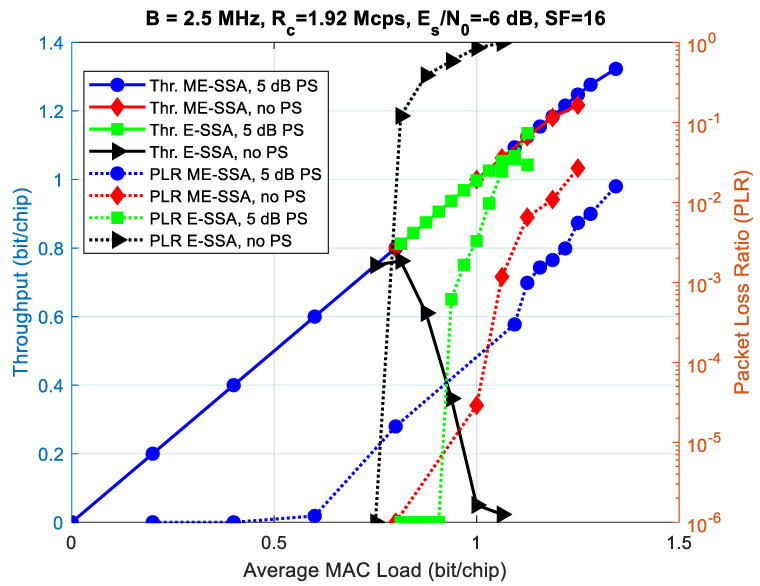
Simulated E-SSA and ME-SSA throughput and PLR performance with and without power randomization (PR), 3GPP FEC coding rate 1/3 with block size of 1200 bits, spreading factor of 16 and E_s_/N_0_ = 6 dB.

**Figure 18 sensors-21-04290-f018:**
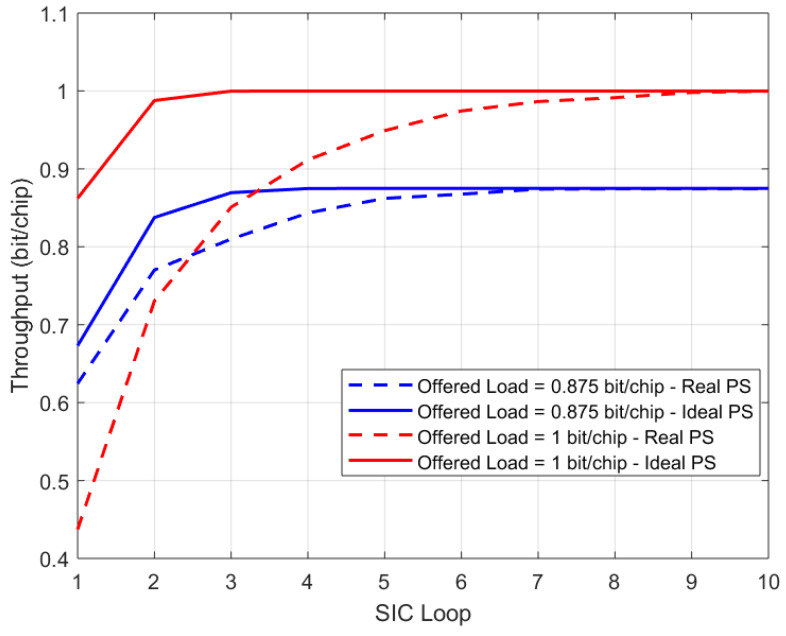
Simulated ME-SSA throughput performance without PR, 3GPP FEC coding rate 1/3 with block size of 1200 bits, spreading factor of 16 and E_s_/N_0_ = 6 dB, for both ideal and actual demodulator preamble searchers (PSs).

**Figure 19 sensors-21-04290-f019:**
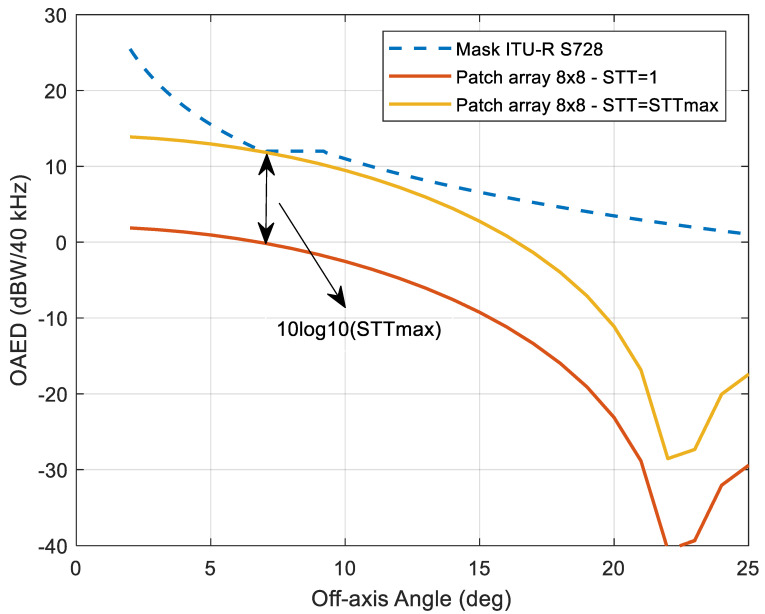
UC#3 OAED constraint.

**Figure 20 sensors-21-04290-f020:**
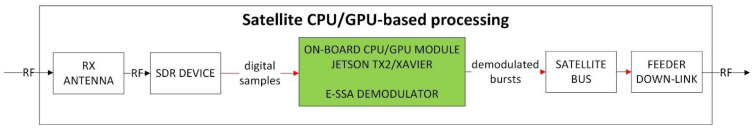
The satellite CPU/GPU payload implementing on-board E-SSA demodulator.

**Figure 21 sensors-21-04290-f021:**
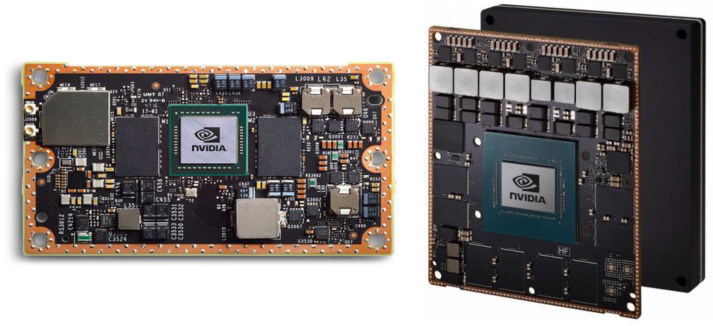
The Jetson TX2 module (credit card size) (**left**) and Jetson AGX Xavier (**right**).

**Figure 22 sensors-21-04290-f022:**
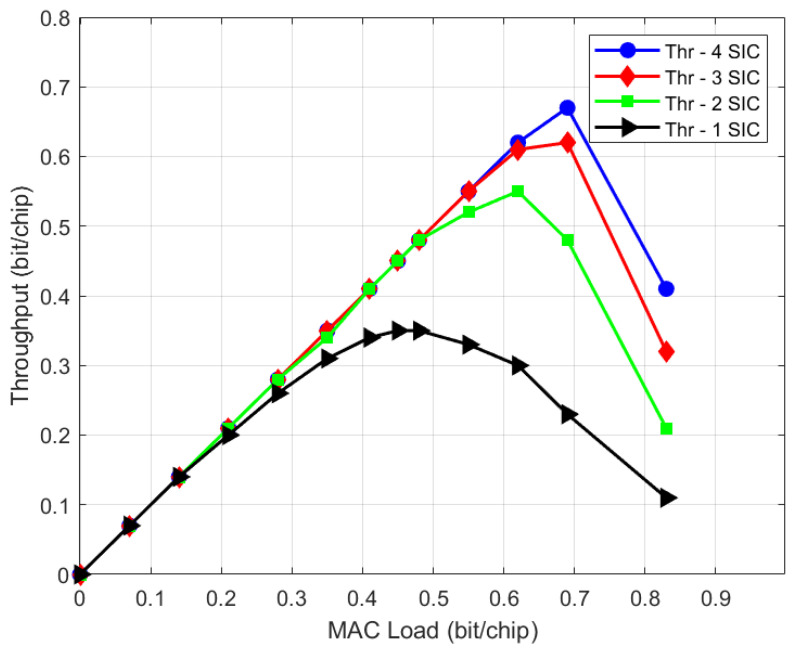
UC#3 IURA throughput performance per SIC iterations.

**Figure 23 sensors-21-04290-f023:**
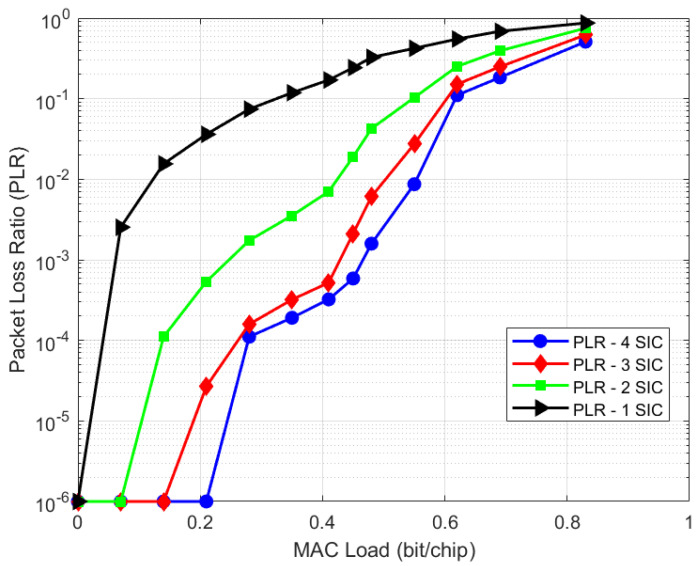
UC#3 IURA PLR performance per SIC iterations.

**Table 1 sensors-21-04290-t001:** UC#1: F-SIM results.

	F-SIM Cr3840Sf16Ds38		F-SIM Cr3840Sf64Ds38	
	AWGN	AWGN + PN	AWGN	AWGN + PN
Throughput (bit/chip)	0.67	0.67	1.46	1.44
PLR	0.001	0.001	0.001	0.001
Throughput (bit/chip)	0.7	0.7	1.59	1.53
PLR	0.01	0.01	0.01	0.01
Throughput (bit/chip)	0.95	0.86	1.71	1.62
PLR	0.1	0.1	0.1	0.1

**Table 2 sensors-21-04290-t002:** UC#2 performance results.

IURA Cr220Sf256Ds38 Performance (AWGN+PN) with OAED Constraint			
Per terminal bit rate (bit/s)	STTmax	Aggregate throughput (bit/chip)	PLR
28	15	0.02	10^−6^

**Table 3 sensors-21-04290-t003:** Non-optimized demodulation performance for GPU device usable on small LEO satellite.

Type of Board	Modcod	Bandwidth	Doppler Shift	Measured Performance
Jetson TX2	FSIM 1920 Kchip/s, SF 128, size 150 bytes	2.5 MHz	No	55 Kbps, 45 pkt/s, 1 iSIC
Xavier	FSIM-like 168 Kchip/s, SF 16, size 38 bytes	200 KHz	Yes, LEO like	7 Kbps, 23 pkt/s, 2 iSICs

**Table 4 sensors-21-04290-t004:** UC#3: IURA results.

	IURA Cr220Sf16Ds38			
	0 SIC	1 SIC	2 SIC	3 SIC
Throughput (bit/chip)	0.06	0.25	0.43	0.465
PLR	0.001	0.001	0.001	0.001
Throughput (bit/chip)	0.12	0.42	0.5	0.55
PLR	0.01	0.01	0.01	0.01

## Data Availability

Not applicable.

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
