# Peer review of "Is Satellite Ahead of Terrestrial in Deploying NOMA for Massive Machine-Type Communications?"

_sensors, 2021, doi:10.3390/s21134290_

Round 1

Reviewer 1 Report

This paper introduced the application of non-orthogonal multiple access (NOMA) techniques in satellite networks. The NOMA-based S-MIM and F-SIM systems were described in detail, and the throughput and packet loss rate performance for F-SIM system were showed in a typical GEO Ku-band scenario with a SmartLNB terminal. Moreover, this paper provided an overview of the developed F-SIM system ground segment elements, such as the user terminals and the gateway station. Several ongoing research and development activities aiming to improve the system performance and to expand the use cases were also summarized. However, there are several major concerns to be addressed.

- This paper outlined the NOMA-based satellite network, the most recent ground segment elements and some ongoing projects. However, this paper mainly focuses on the introduction of related hardware. It will be better to include the architectures, techniques, and challenges of NOMA-based satellite communication networks. For example, the imperfect channel state information will degrade the superiority of the NOMA scheme. How is channel estimation integrated with the applying of the NOMA scheme in realistic scenarios?

- To achieve high power efficiency, the high power amplifier is usually driven at its saturation point in satellite communications. Therefore, the nonlinear distortions resulted from the high power amplifier may degrade the performance of communication systems. In practical NOMA-based satellite communication systems, how to mitigate the effect of the nonlinear distortions?

- There are still some typos in this article, such as the word “case of case of” on page 16. Please correct these typos.

Reviewer 2 Report

This review paper discusses the application of NOMA into satellite systems. Basically, it provides some interesting discussion and is tutorial in nature. My comments are given as:

  1. The presentation of this work needs to be significantly improved. For example, the font in  figs is too small, and resolution of figs is low.
  2. A brief discussion of curve trend in Figs. 1 and 4 is necessary.
  3. It is better to add a short introduction of recent interest of space-air-ground integration, and may consider include relevant literature, such as: Joint UAV hovering altitude and power control for space-air-ground IoT networks, IEEE Internet of Things Journal
  4.  Please consider giving more room for the future work discussion.

Round 2

Reviewer 2 Report

I have no further comment to the authors.